# Heat Strain and Use of Heat Mitigation Strategies among COVID-19 Healthcare Workers Wearing Personal Protective Equipment—A Retrospective Study

**DOI:** 10.3390/ijerph19031905

**Published:** 2022-02-08

**Authors:** Coen C. W. G. Bongers, Johannus Q. de Korte, Mike Zwartkruis, Koen Levels, Boris R. M. Kingma, Thijs M. H. Eijsvogels

**Affiliations:** 1Department of Physiology, Radboud Institute for Health Sciences, Radboud University Medical Center, Philips van Leijdenlaan 15, 6525 EX Nijmegen, The Netherlands; Coen.Bongers@radboudumc.nl (C.C.W.G.B.); Yannick.deKorte@radboudumc.nl (J.Q.d.K.); mikezwartkruis@hotmail.com (M.Z.); 2Institute of Training Medicine and Training Physiology, TGTF, Royal Netherlands Army, Herculeslaan 1, 3584 AB Utrecht, The Netherlands; k.levels@mindef.nl; 3Department of Human Performance, Unit Defence, Safety and Security, TNO, The Netherlands Organization for Applied Sciences, Kampweg 55, 3769 DE Soesterberg, The Netherlands; boris.kingma@tno.nl; 4Department of Energy Technology, Eindhoven University of Technology, Groene Loper 19, 5612 AP Eindhoven, The Netherlands

**Keywords:** thermal stress, heat strain, COVID-19 nurses, health care personnel, cooling interventions, protective clothing

## Abstract

The combination of an exacerbated workload and impermeable nature of the personal protective equipment (PPE) worn by COVID-19 healthcare workers increases heat strain. We aimed to compare the prevalence of heat strain symptoms before (routine care without PPE) versus during the COVID-19 pandemic (COVID-19 care with PPE), identify risk factors associated with experiencing heat strain, and evaluate the access to and use of heat mitigation strategies. Dutch healthcare workers (n = 791) working at COVID-19 wards for ≥1 week, completed an online questionnaire to assess personal characteristics, heat strain symptoms before and during the COVID-19 pandemic, and the access to and use of heat mitigation strategies. Healthcare workers experienced ~25× more often heat strain symptoms during medical duties with PPE (93% of healthcare workers) compared to without PPE (30% of healthcare workers; OR = 25.57 (95% CI = 18.17–35.98)). Female healthcare workers and those with an age <40 years were most affected by heat strain, whereas exposure time and sports activity level were not significantly associated with heat strain prevalence. Cold drinks and ice slurry ingestion were the most frequently used heat mitigation strategies and were available in 63.5% and 30.1% of participants, respectively. Our findings indicate that heat strain is a major challenge for COVID-19 healthcare workers, and heat mitigations strategies are often used to counteract heat strain.

## 1. Introduction

The severe acute respiratory syndrome coronavirus-2 (SARS-CoV-2) and the associated development of coronavirus disease 2019 (COVID-19) place a significant burden on the healthcare system. Healthcare workers are exposed to long working hours and a high workload [1]. This could result in, among other things, mental health problems [2], compromised acute care [3] and an increased number of days-off taken by healthcare workers [4], which might potentially increase the risk of cross-contamination and higher burden at work. To protect healthcare workers from cross-contamination of COVID-19 patients, extensive personal protective equipment (PPE) is worn, such as isolation gowns, hair caps, eye protection, facemask and gloves. The PPE allows, by design, almost no ventilation, which impedes both convective and evaporative heat loss and results in a rise in core temperature and indirectly high sweat rates [1,5,6]. Therefore, the combination of the exacerbated workload and the impermeable nature of the PPE might result in extensive increments in perceived heat strain among healthcare workers [1,7,8].

Heat strain can be classified as physical heat-related symptoms (i.e., thirst, fatigue, (excessive) sweating, uncomfortable warmth) and their effect on work performance (i.e., slower work performance, less accurate execution of work activities). Previous cross-sectional studies reported a high prevalence of heat strain symptoms among COVID-19 healthcare workers. For example, a survey distributed among British healthcare workers (n = 224) revealed that 93% of the respondents experienced multiple heat strain symptoms (i.e., thermal discomfort, headache, fatigue, excessive sweating), whereas 65% and 76% experienced that heat stress impaired their cognitive and physical performance, respectively [5]. Similar observations were made in healthcare workers (n = 356) from Singapore, India and Italy, as thirst, excessive sweating, exhaustion, and desire to go to comfort zones were often reported [6,7]. An important limitation of these studies is the lack of insight into the prevalence of these complaints in the pre-COVID era (i.e., during similar medical duties but without PPE), which hampers the interpretation of the data.

To reduce heat strain in the workplace, heat mitigation strategies can be applied prior to (pre-cooling), during (per-cooling) and after (post-cooling) work shifts [8,9,10]. These mitigation strategies are based on their use in athletes, firefighters, and military personnel, and have been proven to effectively improve physiological and perceptual responses [11,12,13]. We have previously shown that translating these concepts from sports science to medical practice (i.e., a phase change material cooling vest) can significantly improve thermal comfort among healthcare workers working at COVID-19 wards wearing PPE [14]. However, it is not yet known to what extent heat mitigation measures are available for healthcare workers in clinical practice and how often these strategies are typically used.

Therefore, the aims of the present study were to (1) compare the prevalence of heat strain symptoms in healthcare workers before (routine care without PPE) versus during the COVID-19 pandemic (COVID-19 care with PPE), (2) identify healthcare worker subgroups experiencing more heat strain while wearing PPE during COVID-19 care, and (3) evaluate the access to, and use of, heat mitigation strategies.

## 2. Materials and Methods

### 2.1. Study Population

Dutch healthcare workers (i.e., physicians, nurses, nursing assistants, paramedics, nurse practitioners and physician assistants) were invited to participate in this study. Participants were eligible if they worked in the past three months at a general, medium, or intensive care COVID-19 unit for at least 1 week whilst wearing PPE. Participants had to wear PPE that consisted of at least isolation gowns, eye protection, and facemasks. Hair caps were not part of the standard PPE in Dutch hospitals and were therefore not listed as inclusion criteria. Participants provided informed consent prior to study participation, and the study received approval (#2020-6379) from the local Medical Ethical Committee of the Radboud university medical center.

### 2.2. Study Protocol

We used an open recruitment strategy using an online link that was distributed to our professional network of Dutch hospitals, as well as using advertisements on social media (Facebook, LinkedIn and Twitter) and a national newspaper. Participants who fit the in- and exclusion criteria were allowed to complete the questionnaire and participate in this study. Participants were invited to complete an online questionnaire between July and October 2020. Within this timeframe, the Netherlands was a moderate-to-high risk region, facing the first wave and start of the second wave of COVID-19 infections. The questionnaire was co-developed with healthcare workers working in COVID-19 care and contained multiple-choice questions, attitude scales and open questions. Furthermore, we allowed participants to provide additional information about their experience of wearing PPE and the use of heat mitigation strategies.

The online questionnaire consisted of three domains: (1) personal and work characteristics, (2) heat strain symptoms before and during the COVID-19 pandemic, and (3) accessibility and usage of heat mitigation strategies. In the first domain, we gathered information about personal (i.e., sex, age, height, weight, sports activity) and work (i.e., type of hospital, type of employment, type of work) characteristics, as well as exposure time at the COVID-19 ward (number of weeks and number of hours). In the second domain, we asked participants about the frequency of heat strain symptoms during routine clinical care without PPE before the COVID-19 pandemic and similar medical duties whilst wearing PPE during the pandemic. These heat strain symptoms were related to physical heat-related symptoms (i.e., agitation/irritability, shortness of breath, decrease in concentration, dizziness/light-headedness/faint, thirst, headache, nausea, fatigue, (excessive) sweating, uncomfortable warmth) and their effect on work performance (i.e., stop work earlier, slower work performance, less accurate execution of work activities). The third domain focused on the accessibility and usage of heat mitigation strategies. We asked participants which heat mitigation strategies, if any, were available at their ward. We specifically focused on: ice-slurry ingestion, cold drinks, cooling vest, cold towel, ice cream, take an earlier or longer break, and/or stay in a cooler room during breaks. Furthermore, we asked participants how often they used those mitigation strategies on a weekly basis.

### 2.3. Data Analysis

Analyses were performed using SPSS version 25 (IBM Corporation, Armonk, NY, USA) and the level of significance was set at *p* < 0.05. Data were statistically (Shapiro–Wilk test) and visually checked for normality. Descriptive data were presented as mean ± standard deviation or median [interquartile range], and frequencies were reported as percentages. The McNemar’s test was used to assess differences in prevalence of heat strain symptoms experienced during medical duties before and during the COVID-19 pandemic. Furthermore, odds ratios (OR), including 95% confidence intervals, were calculated using a Chi-Square test to examine the impact of sex (male versus female), age (<40 years versus ≥40 years), sports activity level (0–3 h per week versus >3 h per week) and exposure time (<235 h versus ≥235 h) on the prevalence of heat strain symptoms during COVID-19 care while wearing PPE, as these parameters were previously shown to impact human thermoregulation [15]. Exposure time was calculated as the product of the number of weeks worked on COVID-19 wards and the number of hours per week. The age groups were was based on a previous suggested onset of age-related decline in thermoregulation [16], while the groups for sports activity level and exposure time were based on the frequency distribution for sports activity level (0–3 h per week = 76.7% vs. >3 h per week = 23.3%) and exposure time (<235 h = 49.9% and ≥235 h = 50.1%). Moreover, a multivariate logistic backward regression analysis was used to examine the overall impact of sex, age, sports activity level and exposure time on the occurrence of any heat strain symptoms. Additionally, interaction terms for sex, age, sports activity level and exposure time were added to the regression analysis as well. Variables with a *p*-value < 0.10 were included in the multivariate logistic regression model.

## 3. Results

A total of 852 questionnaires were collected from 17 July to 25 October 2020, from which 61 participants (7%) were excluded due to hospital location (non-Dutch, n = 2), department type (i.e., nursing home and home care, n = 51), and type of work activities (i.e., general practitioners and employees of COVID-19 test facilities, n = 8). Hence, the final cohort that were used in the subsequent data analyses was consisting of 791 participants. From which the large majority of healthcare workers worked as a nurse or nursing assistant on a COVID-19 ward (88%).

### 3.1. Participant Characteristics

The median age of the participants was 32 [27–45] years (range: 18–66 years), and the majority were female (n = 683, 86%; Table 1). Moreover, 65.9% (n = 521) and 34.1% (n = 270) of the participants were <40 years and ≥40 years, respectively. Participants worked at a COVID-19 nursing ward (n = 406, 51.3%), medium to intensive care (n = 328, 41.5%), first aid or emergency care (n = 26; 3.3%), or other clinical departments (i.e., rehabilitation, radiology, operating room (n = 31; 3.9%)). Participants worked on average 10 [6–10] weeks for 28 [24–28] hours per week at a COVID-19 unit, with a median cumulative exposure time of 235 [141–280] hours.

### 3.2. Prevalence of Heat Strain Symptoms

Overall, 93% of the participants experienced at least one heat strain symptom while performing COVID-19 medical duties with PPE. In contrast, only 30% of the participants experienced heat strain symptoms during similar work activities without PPE before the COVID-19 pandemic. Thus, participants experienced heat strain symptoms more often during medical duties with PPE versus without PPE (OR = 25.57 (95% CI = 18.17–35.98)). The most prevalent symptoms while wearing PPE were thirst (79%), thermal discomfort (78%), excessive sweating (68%), fatigue (60%), headache (60%) and shortness of breath (55%, Figure 1A). Moreover, 12%, 33% and 19% of healthcare workers reported that they stop work earlier, and that they perform their medical duties slower and less accurately, respectively, while wearing PPE. All symptoms were more prevalent while performing COVID-19 medical duties in PPE compared to similar work activities before the COVID-19 pandemic without PPE (all *p*-values < 0.01; Figure 1A).

When participants reported thermal discomfort, excessive sweating, or thirst, 40–60% of the participants experienced those symptoms very often or almost always, while less than 2% of the participants indicated that they rarely experience these symptoms (Figure 1B). Other symptoms that were experienced frequently were fatigue, headache, slower work performance and shortness of breath, in which 20–40% of the participants experienced these symptoms very often or almost always (Figure 1B).

### 3.3. Factors Associated with Heat Strain Symptoms

Female sex was associated with a higher prevalence of excessive sweating (69.1% vs. 59.3%, OR = 1.54 (95% CI = 1.01–2.33)), headache (64.0% vs. 36.1%, OR = 3.14 (95% CI = 2.06–4.80)), fatigue (63.0% vs. 42.6%, OR = 2.29 (95% CI = 1.52–3.46)), shortness of breath (57.0% vs. 43.5%, OR = 1.72 (95% CI = 1.14–2.59)) and dizziness (30.7% vs. 18.5%, OR = 1.95 (95% CI = 1.17–3.26)) while performing COVID-19 medical duties compared to males (Figure 2A). In contrast, females less often reported a loss of work accuracy compared to males (26.9% vs. 18.0%, OR = 0.60 (95% CI = 0.38–0.96)). Age <40 years was associated with a higher prevalence of excessive sweating (70.4% vs. 62.6%, OR = 1.50 (95% CI = 1.12–2.02)), headache (63.9% vs. 53.0%, OR = 1.67 (95% CI = 1.25–2.22)), shortness of breath (57.8% vs. 50.0%, OR = 1.41 (95% CI = 1.06–1.87)), dizziness (34.2% vs. 19.3%, OR = 2.25 (95% CI = 1.60–3.16)), stop work earlier (13.6% vs. 8.9%, OR = 1.83 (95% CI = 1.14–2.93)) and nausea (6.9% vs. 4.1%, OR = 2.16 (95% CI = 1.13–4.14)) (Figure 2B). Sports activity in daily life did not impact the prevalence of heat strain symptoms (Figure 2C), but participants with a high PPE exposure time experienced more often headache (65.5% vs. 54.9%, OR = 1.55 (95% CI = 1.16–2.07)), fatigue (64.4% vs. 55.9%, OR = 1.42 (95% CI = 1.07–1.90)), a decreased concentration (42.4% vs. 34.4%, OR = 1.40 (95% CI = 1.05–1.87)) and agitation (31.8% vs. 22.5%, OR = 1.60 (95% CI = 1.17–2.20)) compared to peers with a low exposure time (Figure 2D). Our multivariate logistic regression analysis confirmed our univariate outcomes as sex (*p* = 0.001, OR = 0.32 (95% CI = 0.16–0.65)) and age (*p* = 0.049, OR = 0.53 (95% CI = 0.28–0.996)) were retained in the final model to predict heat strain symptoms among COVID-19 healthcare workers. Furthermore, we found no interaction between age, sex, sports activity level and exposure time in our multivariate logistic regression (all *p*-values > 0.10).

### 3.4. Heat Mitigation Strategies

87% of the participants had access to at least one heat-mitigation strategy at their ward. The most frequent available interventions were cold drinks (66%), ice slurry (41%), ice cream (30%) and cooling vests (25%; Figure 3A). However, not all heat-mitigation strategies were used with a similar frequency. For example, cold drinks (82%), a cold room (76%), longer breaks (48%), more breaks (29%) and ice slurry ingestion (29%), were typically used ≥4 times per week if they were available, whereas cooling vests and cold towels were used less often (Figure 3B).

## 4. Discussion

The aims of this study were to (1) compare the prevalence of heat strain symptoms in healthcare workers during routine clinical care versus COVID-19 care, (2) identify healthcare workers subgroups experiencing a greater heat strain while wearing PPE during COVID-19 care, and (3) evaluate the access to and use of heat mitigation strategies. We found that the odds to experience heat strain symptoms in healthcare workers was ~25× greater while performing medical duties with PPE compared to without PPE, in which thirst, thermal discomfort, excessive sweating, fatigue, headache and shortness of breath were the most prevalent symptoms. Female healthcare workers and those with an age <40 years were most affected by heat strain, whereas exposure time and sports activity level were not significantly associated with heat strain prevalence. Finally, we demonstrated that 87% of healthcare workers had access to at least one heat mitigation strategy, in which cold drinks or ice slurry were most frequently used by healthcare workers. In aggregate, these findings indicate that heat strain is a major challenge for COVID-19 healthcare workers (i.e., perceived heat strain symptoms and reduced work performance) and heat mitigations strategies are used to counteract the additional burden for healthcare workers.

A meta-analysis indicated that occupational heat strain symptoms were four times more likely to occur during a single work shift under heat stress (ambient temperature >37 °C) compared to thermoneutral conditions [17]. Although it can be assumed that ambient temperatures are well controlled and considered thermoneutral at COVID-19 wards, this is not the case for the insulated micro-environment that is created underneath PPE. We previously demonstrated that the average ambient temperature on two COVID-19 wards in the Netherlands was 23 °C, whereas sub-PPE temperatures can increase up to ~36 °C [14]. Moreover, it has been reported that the trapped sub-PPE air contains more water vapor relative to the surrounded air [18], leading to a lower evaporative capacity and a greater thermoregulatory challenge even in thermoneutral conditions. It is therefore no surprise that almost all healthcare workers (93% of healthcare workers) experienced heat strain symptoms during COVID-19 medical duties with PPE, and that this prevalence was significantly higher compared to similar medical duties without PPE (30% of healthcare workers). These findings reinforce initial observations from UK, Italy, Singapore and India [5,6,7] and emphasize the magnitude of PPE-induced heat strain among healthcare workers.

Females more often experienced excessive sweating, fatigue, headache, shortness of breath and dizziness during their work, while they less often reported a lower work accuracy compared to males. The sex-specific higher prevalence of heat strain symptoms is in accordance with a previous study that concluded that females are at a thermoregulatory disadvantage compared to males when wearing PPE while exercising in a hot environment [19]. Next to the impact of sex, healthcare workers <40 years more often reported heat strain symptoms compared to their older peers. This is in contrast to our hypothesis as one might expect that older healthcare workers have a higher prevalence of heat strain symptoms due to the age-related decline in thermoregulatory function (i.e., reduced vasomotor control, lower thermal sensitivity and reduced evaporative capacity) [20,21]. Sports activity level in daily life did not impact the prevalence of heat strain symptoms among healthcare workers, which suggests that having a physically active lifestyle, and therefore a thermoregulatory system that is more often exposed to increments in core temperature, does not reduce heat strain during work. In short, female healthcare workers and those with an age <40 years were most affected by heat strain during medical duties in PPE and could benefit more from heat mitigation strategies. Hospital managers of COVID-19 wards can use this information to offer heat mitigation measures to healthcare workers that have a greater risk of heat strain.

Occupational heat stress can impact health (i.e., heat-related injuries that may progress to heatstroke or death), safety (i.e., psychophysical strain such as discomfort, fatigue and coordination loss) and work productivity [22]. Our study demonstrated that healthcare workers not only reported heat strain symptoms that relate to health (i.e., discomfort, headache, thirst, nausea), but also symptoms that relate to the execution of their work activities (i.e., reduced work speed, less accurate work, loss of coordination). This is alarming, as these symptoms could increase the risk of work-related injuries (i.e., health, recovery or overall outcome) in both healthcare workers and their patients [17,23,24,25]. Furthermore, it has been suggested that heat strain might increase the risk of self-contamination by healthcare workers during the post-work shift PPE removal process [23]. This could be explained by (1) the limiting work shift duration, and thereby increased frequency of PPE removal and (2) greater chance of a misstep during the removal process due to a decreased cognitive function [23]. During an already intensive and stressful crisis like the current COVID-19 pandemic, comfortable working conditions are of utmost importance, and therefore, healthcare workers and their managers need to apply heat mitigation measures to reduce the prevalence of heat strain symptoms and lower injury risk.

Heat mitigation strategies were accessible to 87% of healthcare workers, in which 66% and 41% of healthcare workers had access to cold drinks or ice slurry, respectively. This is the first study that evaluated the access to and use of heat mitigation strategies for healthcare workers. The availability of heat mitigation strategies for healthcare workers may depend on the financial resources and the occupancy rate of a hospital, and therefore, the most frequently used cooling strategy could differ per country or region. Remarkably, ~13% of healthcare workers had no access to one of the mitigation measures, highlighting the room for improvement for hospital managers. Second, the most frequently used strategies (≥4 times per week) were drinking cold drinks (82%), going to an airconditioned room (48%), and taking longer breaks (48%). Although cooling vests were available in 28% of the healthcare workers who participated in our study, only 9% of the healthcare workers actually used the cooling vests during work. However, a recent study demonstrated that wearing a cooling vest sub-PPE can significantly improve thermal comfort and lower thermal sensation during COVID-19 medical duties [14]. A potential explanation for the limited use of cooling vests among healthcare workers could be that the number of available cooling vests per COVID-19 ward was limited, and therefore the accessibility was poor. Thus, heat mitigation strategies were widely available among Dutch healthcare workers, from which cold drinks, going to a colder room and taking longer breaks were most frequently reported. Hospital managers could explore how these preferred strategies and/or alternative strategies could be implemented more widely among their personnel. This is important because it has been suggested that cold water/ice slurry ingestion, cooling of the hands in running water or ice water (even while wearing protective gloves) and using a cooling/ice are effective in reducing heat strain [26]. Alternatively, heat mitigation measures were often used as pre-cooling strategies, which might not be sufficient to remain effective throughout a complete work shift of ~3 h. Literature suggested that more aggressive cooling strategies that can be used during work/exercise have a greater cooling potential and could therefore be more effective to reduce heat strain [9].

Heat mitigations strategies can also focus on reducing exposure time. In our study, healthcare workers more often took longer breaks during, and in between, their work shifts to deal with the work and PPE-induced heat stress. Previous studies demonstrated that applying a 3:1 work-rest ratio can markedly decrease thermal strain during moderate intensity work (according to the definition of the American Conference of Governmental Industrial Hygienists), with a greater impact for older (58 ± 5 years) versus younger employees (21 ± 3 years) [27,28]. Additionally, it has been demonstrated that the adverse heat strain effects of PPE (i.e., thirst, exhaustion, headache) were associated with longer work shift durations [29]. Therefore, work shifts of healthcare workers, which can be classified as moderate intensity as well, should be interrupted by more and longer breaks to reduce heat strain, which emphasizes the necessity of measures to prevent self-contamination. Potentially, this could improve physical and cognitive performance and thereby reduce the risk of work-related injuries and contamination for both the patient as the healthcare workers.

The strengths of our study are the large sample size, the comparison of heat strain symptoms while performing medical duties with and without PPE, the identifications of groups at risk, and the exploration of heat mitigations strategies available to COVID-19 healthcare workers. However, some limitations should be taken into account. First, all the questionnaires have been taken during summer/autumn in the Netherlands (July to October 2020). Although we have no information about the acclimatization status of the participants, we can assume that healthcare workers were at least partly acclimated as the Netherlands was hitting a 13-day heatwave in August 2020. Therefore, one might suggest that the prevalence of heat strain symptoms is even higher in unacclimated healthcare workers. Second, 86% of the healthcare workers were female, which suggests that males might be underrepresented in our cohort. However, this ratio between the number of males and females included in our study correctly reflects the ratio currently seen on nursing wards in the Netherlands (87% female) [30]. Third, due to the retrospective study design, we were not able to demonstrate whether there was a causal relationship between the use of heat mitigation strategies and the prevalence of heat strain symptoms during medical duties in PPE. Future field studies should focus on the effectiveness of heat mitigation strategies. Fourth, there is a potential risk for selection bias in our study, as one might suggest that only healthcare workers who experienced heat strain symptoms have completed the online questionnaire. However, in the information package for the participants, we describe that our study is aimed to describe the workload of healthcare workers during COVID-19 medical duties without specific emphasis on heat strain. Therefore, we assume that selection bias was limited and that the results of this study are generalizable to healthcare workers working in PPE in general.

## 5. Conclusions

The prevalence of heat strain symptoms in healthcare workers was ~25× greater while performing medical duties with PPE compared to without PPE, in which thirst, thermal discomfort, excessive sweating, fatigue, headache and shortness of breath were the most prevalent symptoms. Moreover, we demonstrated that heat strain could impact healthcare workers performance as well. Female healthcare workers and those with an age <40 years were most affected by heat strain, whereas exposure time and sports activity level were not significantly associated with heat strain prevalence. Heat mitigation measures were available to 87% of healthcare workers, but the accessibility and frequency of use differed largely between individuals and hospitals. Cold fluid ingestion and recovery in an airconditioned room were most often used when available, but despite the use of these mitigation strategies, heat strain was experienced by most of the healthcare workers. In aggregate, hospital managers of COVID-19 wards can use this information to offer heat mitigation measures to healthcare workers that have a greater risk of heat strain to reduce the prevalence of heat strain symptoms and lower injury risk.

## Figures and Tables

**Figure 1 ijerph-19-01905-f001:**
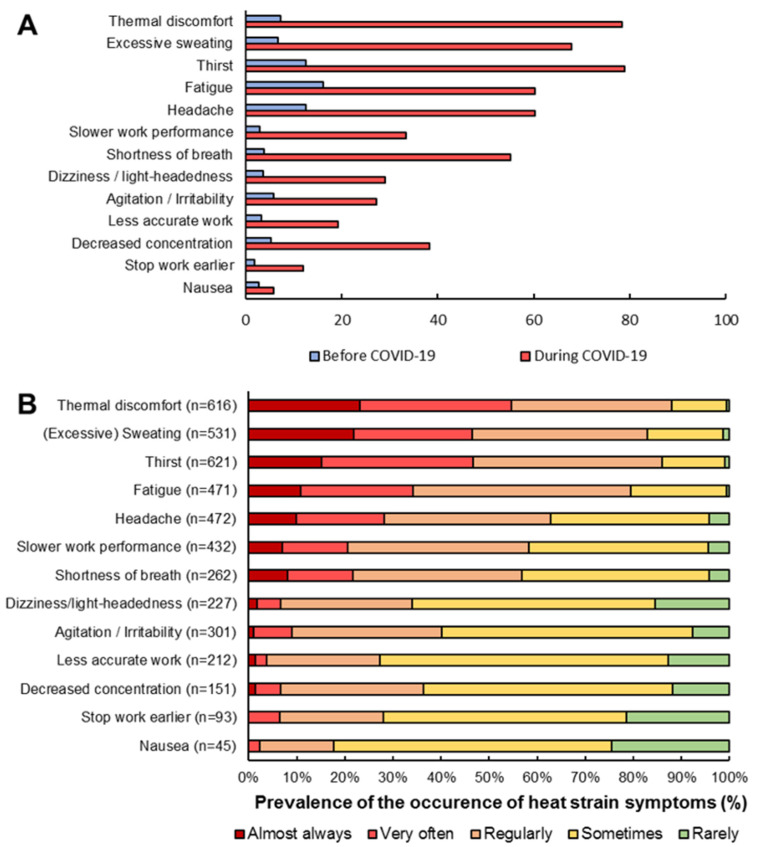
(**A**) Prevalence of heat strain symptoms among healthcare workers during routine care (blue bars, without PPE) and COVID-19 care (red bars, with PPE). (**B**) Prevalence of the occurrence of heat strain symptoms. Large variability is observed in how often specific symptoms were reported. For example, the majority (54.8%) of healthcare workers who experienced thermal discomfort had this very often to always during their work shift, whereas this was only 2.2% for nausea.

**Figure 2 ijerph-19-01905-f002:**
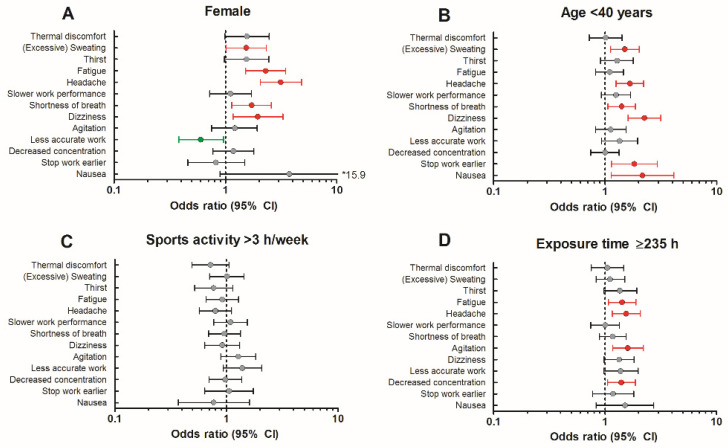
Impact of female sex (**A**), age < 40 years (**B**), sports activity level > 3 h/week (**C**) and PPE exposure time ≥ 235 h (**D**) on the odds to experience heat strain symptoms during COVID-19 care while wearing PPE. Red, grey and green dots represent higher, similar and lower odds, respectively, for any given heat strain symptom.

**Figure 3 ijerph-19-01905-f003:**
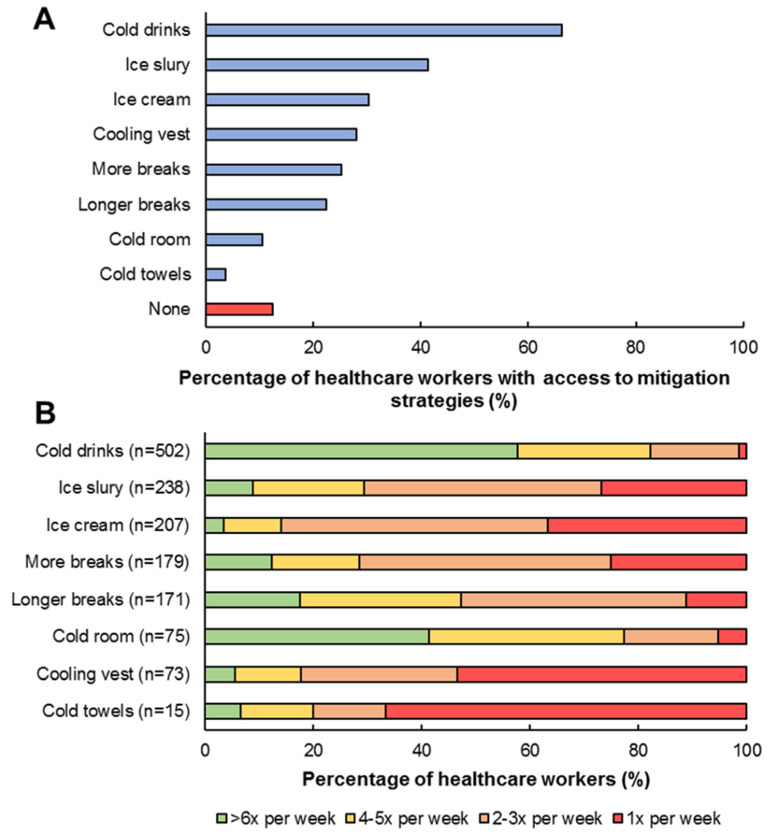
Prevalence of available heat mitigation strategies for healthcare workers involved in COVID-19 care (**A**) and how often these countermeasures were applied (**B**).

**Table 1 ijerph-19-01905-t001:** Participant characteristics.

Participant Characteristics	Total Group (n = 791)	Male (n = 108)	Female (n = 683)
Age (years)	32 [27–45]	33 [29–45]	32 [26–45]
Height (cm)	173 ± 8	183 ± 8	171 ± 7
Weight (kg)	71.0 [64.0–80.0]	82.0 [75.0–90.8]	70.0 [63.0–79.0]
BMI (kg/m^2^)	23.8 [21.6–26.7]	24.5 [22.2–26.3]	23.7 [21.5–26.8]
Sports activity level (hours per week)			
<1 h (n(%))	170 (21.5%)	19 (17.6%)	151 (22.1%)
1–3 h (n(%))	437 (55.2%)	48 (44.4%)	389 (57.0%)
4–6 h (n(%))	164 (20.7%)	32 (29.6%)	132 (19.3%)
≥7 h (n(%))	20 (2.5%)	9 (8.3%)	11 (1.6%)
Type of work			
Medium care/Intensive care (n(%))	328 (41.5%)	62 (57.4%)	266 (38.9%)
Nursing ward (n(%))	406 (51.3%)	35 (32.4%)	371 (54.3%)
First aid/Emergency care (n(%))	26 (3.3%)	5 (4.6%)	21 (3.1%)
Other medical departments (n(%))	31 (3.9%)	6 (5.6%)	25 (3.7%)
Number of weeks at COVID-19 ward	10 [6–10]	10 [6–10]	10 [6–10]
Hours per week at COVID-19 ward	28 [24–28]	28 [24–28]	28 [24–28]
Exposure time (hours)	235 [141–280]	209 [168–280]	235 [141–280]

Data are presented as mean ± SD, median [interquartile range] or frequency (%).

## Data Availability

The data presented in this study are available on request from the corresponding author.

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
