# Peer review of "Heat Strain and Use of Heat Mitigation Strategies among COVID-19 Healthcare Workers Wearing Personal Protective Equipment—A Retrospective Study"

_ijerph, 2022, doi:10.3390/ijerph19031905_

Round 1

Reviewer 1 Report

The manuscript “Heat strain and use of heat mitigation strategies among COVID-19 healthcare workers wearing personal protective equipment” addresses an interesting topic, it has three objectives:  strain symptoms in HCW before (routine care without PPE) versus during the COVID-19 71 pandemic (COVID-19 care with PPE), 2) identify HCW subgroups experiencing more heat strain while wearing PPE during COVID-19 care, and 3) evaluate the access to, and use of, heat mitigation strategies.  This manuscript requires improvements in the methods and results sections before it can be published.

Introduction

The study was performed between July and October 2020, it would be helpful to learn about

the pandemic stage during this period in the study region, if not available in the country.

Methods

In the method section, the subtitle “Experimental protocol” is used. This information could be misleading because the study design is cross-sectional. This term is used frequently in other designs such as in clinical trials.

The participants came from different sources, such as the professional network of Dutch hospitals as well as using advertisements on social media and in a national newspaper. How many participants came from each source? How many received the questionnaire, and how many answered it? Additionally, was there any strategy applied to verify the validity of the questionnaire?

In the statistical analysis, it is mentioned that normality of the variables distribution was checked visually, why was a statistical test for normality not applied? Were interaction terms tested in the model? Add hypothesis test level of statistical significance.

Results

Health workers invited to participate in the study, encompassed physicians, nurses, nursing assistants, paramedics, nurse practitioners, and physician assistants. It would be useful for further preventive programs to learn if the specific setting's role influences the effect of heat strain. There is no information on the composition of the sample in this regard.

The variable exposure time was dichotomized at 235 hours in length, and it was close to rich statistical significance in the regression analysis. Were other cut-off points analyzed?

The manuscript presents the p-values of the results of the multivariate logistic regression model. Adding a table with the model results including OR and 95%CI will complement the results, also including interaction term if significant in the model.  Interaction effects make the model more complex, but if found it is critical to incorporate it in the model because it provides a more realistic explanation of the predicted variable.

Line 173, HCP provide definition of this acronym

Discussion

Line 222. “Females, aged <40 years with a longer exposure time were most affected by heat strain”. This conclusion is not fully supported by the multivariate logistic regression model results, since the variable exposure time presented a p-value = 0.10.  Maybe adding an interaction term in the model will modify this result.

It is not clear why younger HCW had a higher prevalence of heat strain symptoms. Maybe including a setting role variable (physicians, nurses, nursing assistants, paramedics, etc.) in the model will help to elucidate this association, considering that work intensity is different professions in the HCW group. This may be a confounding variable.

Line 328 “However, this ratio between the number of males and females included in our study correctly reflects the ratio currently seen on nursing wards”. Add references.

Among the limitations of the study, the possibility of selection bias should be considered, also more information about the generalizability of the study results would be useful.

Reviewer 2 Report

this study compares the prevalence of heat strain symptoms before and during the COVID-19 pandemic, and identify risk factors associated with experiencing heat strain. 

the study is well designed and scientifically rigorous. Empirical data is informative. 

Key articles are reviewed such as Lee et al (2020), Davey et al (2021) and Messeri et al (2021). so the review is fine.

I only have minor comments. 

Using acronyms here only hinders readability. HCW should be just healthcare workers and PPE should be just personal protective equipment at least in the abstract. Or leave PPE as is but HCW should be really just healthcare workers

Reviewer 3 Report

The strong side of the paper lies in the experiment, which is properly conducted, described and results are presented extensively. However, the weak side of the paper lies in the theoretical background.

The paper lacks systemic literature analysis, which would be a basis for hypotheses tested with the experiment. It is bizarre that scientific article does not even have a section: theoretical background and goes straight into materials and methods. Especially since there is an extensive body of literature concerning (in wide range) the topic of the article. The article should be amended by theoretical background and results of other studies presenting similar issue.

Moreover, paper lacks properly written conclusions section. There are not limitations of research properly presented, there are no future directions of research. 

Major revisions needed: theoretical background part of the article must be written, conclusions must be rewritten.

Reviewer 4 Report

The objective of this paper was to characterize the prevalence of heat strain symptoms among HCWs before COVID-19 and after COVID-19 and assess heat mitigation strategies. The research methods and writings are sound. I have a few minor comments below.

  • Table 1: Column "heat strain symptoms". I suggest changing the name of the column to "participant characteristics". The authors can state that the heat strain symptoms information is presented in Figure 1.
  • I recommend the authors add the survey to the supplemental materials of this manuscript.

Reviewer 5 Report

Thank you for giving me to review your manuscript. This manuscript is interesting and meaningful for considering the association between COVID-19 health care professionals' health conditions/demographics and their working conditions. Regarding the contents, the following revision should be considered for the quality of research.

The title should be more specific regarding study design.

The introduction should include the issue of risk and benefit of infection control measures regarding COVID-19 more in-depth. There is much interventional research regarding COVID-19 infection control facing aging societies. This research should consist of the part referring to the following articles.

- Greenberg, N., et al., Managing mental health challenges healthcare workers face during the covid-19 pandemic. BMJ, 2020. 368: p. m1211.

- Ohta, R., Y. Ryu, and C. Sano, Effects of Implementation of Infection Control Measures against COVID-19 on the Condition of Japanese Rural Nursing Homes. Int J Environ Res Public Health, 2021. 18(11).

- Verhoeven, V., et al., Impact of the COVID-19 pandemic on the core functions of primary care: will the cure be worse than the disease? A qualitative interview study in Flemish GPs. BMJ Open, 2020. 10(6): p. e039674.

The introduction should clearly include this study's research question and rationale, including the advantage. There are many studies regarding this research topic, especially in primary care contexts.

In the sample section of the method, there are no descriptions regarding sample calculation. Therefore, the authors should descript the sample size calculation.

The statistical analysis should be more described. The authors should explain how to deal with each variable, referring to previous studies.

The discussion should describe the limitation of sampling bias and the results' applicability to other settings, and the future investigation in the limitation part, especially regarding the analysis.

Reviewer 6 Report

This work aimed to compare the prevalence of heat strain symptoms before (routine care without PPE) versus during the COVID-19 pandemic (COVID-19 care with PPE), identify risk factors associated with experiencing heat strain and evaluate the access to and use of heat mitigation strategies. There are some comments for the authors to improve its quality.

  1. Why was heat strain selected as the study focus? Please give more explanations.
  2. How to confirm that all participants are HCW in the online questionnaire?
  3. How to develop the questionnaire contents should be clarified. Did the authors develop the contents based on previous studies? If yes, please add references.
  4. What types of PPE were commonly used by HCW? Can any related statistics be provided in the manuscript?
  5. The authors mentioned that “The questionnaire was co-developed with HCW working in COVID-19 care and contained multiple-choice questions, attitude scales, and open questions.”I cannot see the attitude scales and open questions.
  6. Why were sports activity of 3 hours and exposure time of 235 hours set as a cutting point? sports activity used 76.7% which exposure time used 49.9%.
  7. All participants were aged between 27-45. Why there were no participants aged 20-26 and 46-65? This may cause certain biases to the results.
  8. Heat strain should be measured objectively. The retrospective study design cannot measure the heat strain accurately. The authors should consider this point.
  9. More practical implications of this study should be discussed.

Round 2

Reviewer 1 Report

The authors modified the manuscript following the reviewer's suggestions. The new version is more complete, particularly regarding the statistical analysis.

Reviewer 5 Report

The manuscript has been considerably improved. I think that this paper is suited for inclusion in our journal.

Reviewer 6 Report

I have no further comment.